# Structure of a full-length bacterial polysaccharide co-polymerase

Benjamin Wiseman [1✉], Ram Gopal Nitharwal[1,3], Göran Widmalm [2] & Martin Högbom [1✉]

Lipopolysaccharides are important components of the bacterial cell envelope that among other things act as a protective barrier against the environment and toxic molecules such as antibiotics. One of the most widely disseminated pathways of polysaccharide biosynthesis is the inner membrane bound Wzy-dependent pathway. Here we present the 3.0 Å structure of the co-polymerase component of this pathway, WzzB from *E. coli* solved by single-particle cryo-electron microscopy. The overall architecture is octameric and resembles a box jellyfish containing a large bell-shaped periplasmic domain with the 2-helix transmembrane domain from each protomer, positioned 32 Å apart, encircling a large empty transmembrane chamber. This structure also reveals the architecture of the transmembrane domain, including the location of key residues for the Wzz-family of proteins and the Wzy-dependent pathway present in many Gram-negative bacteria, explaining several of the previous biochemical and mutational studies and lays the foundation for future investigations.

[1] Department of Biochemistry and Biophysics, Stockholm University, Stockholm, Sweden. [2] Department of Organic Chemistry, Stockholm University, Stockholm, Sweden. [3] Present address: Department of Biotechnology, School of Interdisciplinary and Applied Sciences, Central University of Haryana, Mahendragarh, Haryana, India. ✉email: bwise@dbb.su.se; hogbom@dbb.su.se

Bacterial cell surface polysaccharides including lipopolysaccharides (LPS) such as the O-antigen and the enterobacterial common antigen are directly involved in many vital biological processes[1], contribute significantly to the cell's integrity and function as a protective barrier to harmful molecules such as antibiotics[2]. Structural components of the LPS can be isolated from a bacterium or synthesized chemically followed by cross-linking or conjugation to carrier proteins to be used as vaccines to combat pathogenic bacteria[3,4]. Recent biotechnological developments include recombinantly glycosylated proteins for the production of glycoengineered bioconjugates containing pertinent structure components to act as protective vaccines[5].

LPS molecules are comprised of a lipid A region, a core oligosaccharide, and a glycan polymer of repeating sugar units referred to as the O-antigen (Oag); the enterobacterial common antigen (ECA) may instead of the Oag be linked as a polysaccharide to the core of the LPS[6]. The different Oags of Gram-negative bacteria are used for serotyping and are built of repeating oligosaccharide units that differ in size and composition depending on the bacterial species; this creates a huge diversity of polysaccharide structures with varying chemical and biophysical properties. Arguably, the widest dissemination of any polysaccharide biosynthesis system is the bacterial Wzy-dependent pathway[7] where the Oag or ECA polymers are synthesized directly at the inner membrane by three integral membrane proteins: a flippase Wzx that flips a lipid-bound oligosaccharide from the inner to the outer leaflet, a polymerase Wzy that polymerize the oligosaccharides into longer polysaccharide chains, and a polysaccharide co-polymerase (PCP) Wzz that modulates the length of the growing polysaccharide chain[8]. Once the polysaccharide molecule is completed, the ligase WaaL attaches it to Lipid A to generate the final LPS that is then shuttled to the outer membrane via the Lpt system[9]. The components of the Wzy-dependent pathway have been speculated to form a large supercomplex[10]; in particular, it has been shown that the polymerase Wzy and co-polymerase Wzz must work in close contact in order to properly modulate the growing polysaccharide chain length[7,11].

Key to the Wzy-dependent pathway is the polysaccharide co-polymerase Wzz that is known to homo-oligomerize into a unique bell-shaped structure[12–14]. Polysaccharide co-polymerases are a large family of membrane proteins that play an essential role in regulating the length distribution of the synthesized polysaccharide chain by interacting with the membrane-bound polymerase Wzy as it assembles the growing Oag or ECA polysaccharide of the LPS. As well as being assembled into bell-shaped structures common features include a membrane topology consisting of an N (TM1) and C (TM2) terminal transmembrane helix, an elongated periplasmic α-helical domain containing a predicted coiled-coil region and conserved sequence motifs such as a Proline-rich segment proximal to TM2 and a glycine-rich segment within TM2[15]. Despite these common features, polysaccharide co-polymerases can be divided into classes based on the chemical nature of the polysaccharide that they produce[15]. Classes 1 and 2 participate in the Wzy-dependent pathway with Class 1 being further subdivided into subgroups 1a (WzzB and FepE proteins) that regulate Oag chain length and 1b (WzzE proteins) that regulate ECA chain length[15]; while class 2 shares the common characteristics of class 1 they are also associated with an additional tyrosine kinase located at their cytoplasmic domain and are responsible for capsular polysaccharides (CPS) biosynthesis[16]. There is also a third class of polysaccharide co-polymerases that are involved in CPS biosynthesis; however, they are less widespread and there have been conflicting reports of their topology[17,18].

There are several high-resolution structures available of the periplasmic domain of the different homologs WzzB, WzzE, and FepE from various organisms[12–14]. However, despite this intense structural analysis high-resolution structural information is lacking for an intact enzyme, particularly for the membrane-embedded region. An attempt at the crystallization of a full-length E. coli protein resulted in only low diffracting crystals without the transmembrane domain being able to be resolved[19]. Similarly, a low resolution, 9 Å, cryo-EM structure of a full-length protein from Salmonella enterica also failed to resolve the transmembrane domain[20]. With that in mind, we determined and herby report the structure of a fully intact example polysaccharide co-polymerase, WzzB, from E. coli by cryo-electron microscopy to an overall resolution of 3.0 Å. The high quality of the map allowed ~95% of the amino acid sequence to be unambiguously built (Table 1), revealing structural details of the transmembrane and cytoplasmic domains as well as a highly conserved proline-rich segment proximal to the C-terminal transmembrane helix. The structure provides a context for understanding the positioning of highly conserved residues, years of mutagenesis experiments and lays the framework for future studies into understanding the membrane-bound Wzy-dependent polysaccharide biosynthesis pathway. This information about a bacterial biosynthesis system would not only be beneficial in the development of treatments against human pathogens, but would also be a step toward the rational design of these complexes allowing the production of tailor-made biopolymers suitable for medical and industrial purposes[21].

## Results

**Oligomeric state and overall architecture.** Analysis of the metal-affinity-purified full-length WzzB polysaccharide co-polymerase complex of E. coli by size-exclusion chromatography (SEC) revealed a small shoulder (population 1) eluting slightly before the main elution peak (population 2) (Supplementary Fig. 1a) corresponding to sizes of 550 and 220 kDa, respectively (based on SEC calibration standards). In order to further assess the size, purity, and stability of the complex the SEC fractions corresponding to the two populations were analyzed using denaturing (SDS-) and Blue-native (BN-) PAGE (Supplementary Fig. 1b,c). Further analysis by SDS-PAGE confirms the presence of WzzB with an apparent molecular mass of ~45 kDa consistent with the monomer. BN-PAGE analysis revealed thick prominent bands ranging from ~950 to ~600 kDa for population 1 and ~600 to ~300 kDa for population 2 (based on the globular native PAGE markers) in agreement with a large oligomeric structure, albeit slightly larger than the estimated sizes based on SEC standards. Despite the apparent large sizes, visualization of the two populations by both negative-staining and cryo-electron microscopy revealed material suitable for single-particle analysis from population 1 only (Supplementary Fig. 1c, d). Despite containing purified WzzB molecules of a size that should be visible by electron microscopy, population 2 contained no observable complexes that could be analyzed using the single-particle technique. However, negative-stain analysis of this population did reveal small particles of varying sizes and shapes that could not be accurately measured. This may suggest that this population contains multiple oligomeric states of WzzB that under these conditions, for whatever reason, do not assemble into the large octameric complexes seen in population 1. This is presumably consistent with previous cross-linking experiments that suggest that Wzz oligomerization is highly dynamic in nature[22–25]. This could also be a consequence of the T7 promoter based system used to overexpress the protein and/or the detergent used to extract the complex from the membrane.

**Table 1 Cryo-EM data collection, refinement, and validation statistics.**

| | Full-length WzzB | | N-terminal truncated WzzB | |
| --- | --- | --- | --- | --- |
| | EMD-4798 PDB 6RBG | EMD-4791 | EMD-11909 | EMD-11908 |
| *Data collection and processing* | | | | |
| Magnification | ×130,000 | ×130,000 | ×96,000 | ×96,000 |
| Voltage (kV) | 300 | 300 | 300 | 300 |
| Electron exposure (e–/Å$^2$) | 52 | 52 | 50 | 50 |
| Defocus range (μm) | 1.4 to 3.2 | 1.4 to 3.2 | 0.8 to 3.2 | 0.8 to 3.2 |
| Pixel size (Å) | 1.06 | 1.06 | 0.85 | 0.85 |
| Symmetry imposed | C8 | C1 | C8 | C1 |
| Initial particle images (no.) | 289,073 | 289,073 | 183,973 | 183,973 |
| Final particle images (no.) | 52,378 | 52,378 | 76,968 | 76,968 |
| Map resolution (Å) | 3.0 | 3.4 | 4.5 | 7.2 |
| FSC threshold | 0.143 | 0.143 | 0.143 | 0.143 |
| Map resolution range (Å) | 30 to 2.3 | 45 to 2.3 | 57 to 2.8 | 15 to 2.8 |
| *Refinement* | | | | |
| Initial model used (PDB code) | N/A | – | – | – |
| Model resolution (Å) | 3.2 | – | – | – |
| FSC threshold | 0.5 | | | |
| Model resolution range (Å) | 4.3 to 2.3 | – | – | – |
| Map sharpening *B* factor (Å$^2$) | 87 | – | – | – |
| Model composition | | | | |
| Non-hydrogen atoms (no.) | 19,336 | – | – | – |
| Protein residues (no.) | 2468 | – | – | – |
| Ligands (no.) | 0 | – | – | – |
| *B* factors (Å$^2$) | | | | |
| Protein | 99.35 | – | – | – |
| Ligand | – | – | – | – |
| R.m.s. deviations | | | | |
| Bond lengths (Å) | 0.005 | – | – | – |
| Bond angles (°) | 1.187 | – | – | – |
| Validation | | | | |
| MolProbity score | 1.13 | – | – | – |
| Clashscore | 3.44 | – | – | – |
| Poor rotamers (%) | 0.38 | – | – | – |
| Ramachandran plot | | | | |
| Favored (%) | 98.04 | – | – | – |
| Allowed (%) | 1.96 | – | – | – |
| Disallowed (%) | 0.00 | – | – | – |

Generally, the overall architecture resembles a box jellyfish, containing a large bell-shaped periplasmic domain similar to what has previously been reported[12,13,19,20] and significantly, 8 transmembrane domains encircling a large transmembrane chamber (Fig. 1). Without the application of symmetry or an approximate/estimated volume for the complex of protomers, the resulting single-particle analysis identified an octameric complex (Fig. 1). The octameric nature of the complex was identified by reference-free ab initio 3D classification and is clearly visible in an unsymmetrized (C1) map. A reconstructed map without the application of symmetry resulted in an overall resolution of 3.4 Å (Supplementary Fig. 2), and an estimation of local resolution suggests that the large bell-shaped periplasmic domain dominates the image alignment process with an overall resolution of 3.3 Å compared to the transmembrane domain at 4.2 Å. Since an octameric complex is clearly visible in the unsymmetrized map, 8-fold (C8) symmetry was applied which led to a marked improvement in overall resolution to 3.0 Å (Fig. 1, Supplementary Fig. 2). This was especially beneficial to the transmembrane domain for which the resolution was improved by 1 Å, allowing the unambiguous construction of an atomic model for this region. Consistent with previous findings[12,19] initial ab initio 3D classification followed by heterogeneous refinement classified ~25% of the WzzB particles as dimers of the octameric complex

in an apparent pseudo D-symmetry (Supplementary Fig. 3). Attempts to refine these dimers with various symmetries applied resulted only in a resolution of 8–10 Å. Further inspection of the raw micrographs revealed a small amount of even higher-order association of octamers (Supplementary Fig. 1d). Subsequent analysis found that an additional 10% of the dimers of octamers were in fact mixed into the set of particles used to generate the final high-resolution volumes as suggested by weak density seen below many of the 2D class averages (Supplementary Fig. 2b). However, additional rounds of refinements with these particles removed had no effect on the final reconstructions. Even a strict filtering of this final set of particles, by removing over 50% of them had, apart from a slight decrease in resolution, no effect on the overall architecture of the reconstruction. This suggests that the overall structure of the octameric complex is the same regardless of its association with the dimers or not. To confirm this, masking and refinement around a single octamer within a dimer (Supplementary Fig. 3) resulted in the reconstruction of an octameric complex with the same structure as that of single octamers. That includes, importantly, the positioning of the helices of the transmembrane domain.

There have been conflicting reports of the exact oligomeric state of this family of proteins. Various reports have suggested trimeric, pentameric, hexameric, and octameric arrangements of WzzB and

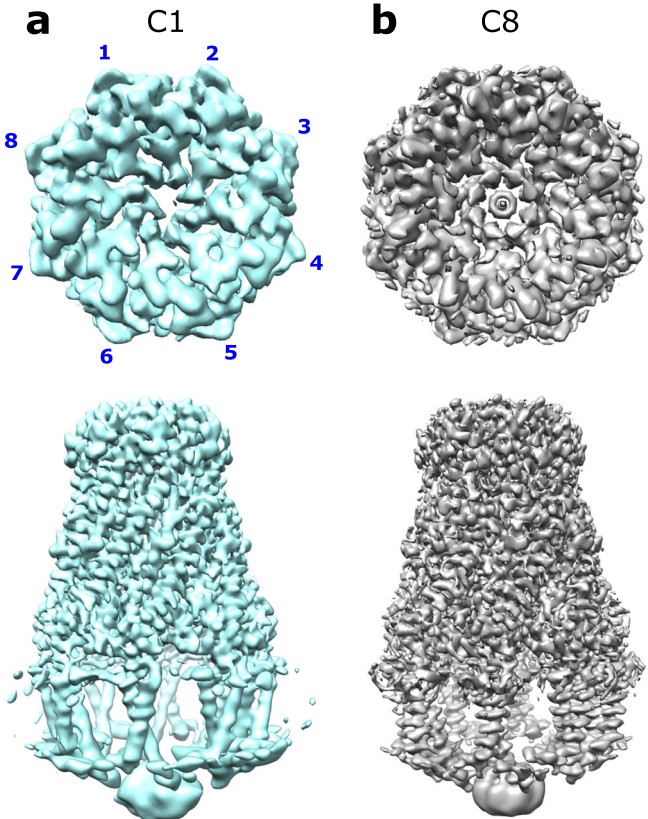

**Fig. 1 Overall structure of *E. coli* WzzB.** Final reconstructed volumes of the full-length WzzB refined without (**a**) and with (**b**) the application of C8 symmetry. Top: view looking down from the top of the periplasmic domain.

WzzE, and a nonameric organization of FepE[12–14,19,25]. Kalynych et al.[19] present convincing structural and biochemical data suggesting that both WzzB and WzzE are octameric in nature and anything smaller than that is an artifact of crystallization. The results presented here are consistent with their findings. This is considerably smaller than estimated by either SEC or BN-PAGE. Based on the larger sizes estimated by SEC and BN-PAGE it has been suggested that these complexes could possibly contain 10–15 protomers per complex[19]. Similarly, our observed SEC elution profiles and native PAGE migration patterns are consistent with very large complexes. However, we believe this to be misleading as our 2D and 3D classifications did not reveal anything larger than an octameric complex. A more likely explanation is the association of two or more complexes via the cytoplasmic domains into dimers, trimers (or even higher order) of octamers that is strong enough to be maintained during BN-PAGE and SEC conditions. Furthermore, a doughnut-shaped DDM detergent micelle protecting the transmembrane domain of each protomer extends the diameter of the transmembrane region to ~135 Å. This large amount of detergent surrounding the transmembrane domain could also account for the large overestimation of size based on SEC and BN-PAGE. The only outlier of this octameric trend is the structure recently determined by single-particle cryo-EM of the WzzB from *Salmonella enterica* sv *Typhimurium* (WzzB[ST]) that is reported as a dodecamer[20], which was previously reported as either a hexamer[25] or an octamer[19], based on size-exclusion-multiangle laser light-scattering and cryo-EM studies of the full-length protein. Also, crystallization of chimeras between this protein and a homolog from *Shigella flexneri*[13] resulted in octameric complexes.

**Periplasmic domain**. The dimensions of the periplasmic domain are very similar to the previously described structures (Fig. 2): a bell-shaped domain ~100 Å in height above the membrane interface that contains a hollow ~70 Å in height interior when measured from the periplasmic membrane interface to the bottom of the L4 loop. It is narrowest at the top of the bell being ~55 Å and widens as it approaches the periplasmic membrane interface to a maximum of ~95 Å in diameter. Similarly, the hollow interior is 30–35 Å in diameter and expands into a large empty chamber ~75 Å in diameter at the periplasmic interface. The bell-shaped complex results from a side-by-side packing of protomers and the interactions observed in this structure are similar to what has been described elsewhere[12,13] with, among others, α1, α7, and α8 of one protomer interacting with the long α6 of an adjacent protomer. We were also able to build a fully intact L4 loop, which is located at the top of the periplasmic domain, ~70 Å away from the membrane interface, and is highly flexible as judged by the majority of the available structures that contain either a distorted or incomplete loop, or is missing the loop all-together in the deposited models[12]. Herein, the loop is positioned on the inside of the cavity and extends to the center of the periplasmic domain (Fig. 2) in a similar conformation as seen in the structure of a *Salmonella typhimurium*/*Shigella flexneri* WzzB chimera[13]. Despite the fact that the location of the loop is far from the membrane interface it is vitally important in the production of very long polysaccharide chains[13,22,26] and Kalynych et al.[26] suggest the flexibility of the L4 loop could mediate conformational changes associated with function. Although the L4 loop does not appear to be required for correct oligomerization[13], multiple residues within this loop are engaged in numerous contacts with residues of neighboring L4 loops at the center of the complex, potentially creating a pore at the top of the periplasmic domain.

The periplasmic domain of this family of proteins has been well characterized and this region of our structure is very similar to the other available structures (Fig. 3) at both the multimeric and protomeric level. Overall, despite a relatively low sequence identity between the three homologs WzzB, WzzE, and FepE[12], the three proteins assemble into remarkably similar bell-shaped structures with hollow interior cavities. This is reflected by the superposition of full multimeric periplasmic domains that crystallized in the correct octameric or nonameric form from different homologs onto our de novo built complex (Fig. 3a). At the protomer level, regardless of the multimeric state that the protein crystalized in, the structural similarity is even more striking with RMDS ranging from 0.8 to 2.8 Å when superimposed (Fig. 3b, c).

**Periplasmic base and linker**. At the periplasmic interface, conserved residues in the linker region connecting the periplasmic domain with the transmembrane domain help form a rigid scaffold that lock each of the transmembrane 2-helix bundles in place (Fig. 4). A proline-rich motif proximal to the C-terminal transmembrane helix encompasses this region. Mutations to alanine residues of the first two prolines of this motif (P284 and P287) had no effect on activity[23]; however, mutation of the third proline (P293) to an alanine[23] or the insertion of a 5-amino acid stretch just prior to this proline[22] resulted in complete loss of function and had a detrimental effect on expression levels. While the first two prolines of this motif are seen in the *trans* conformation, P293 is best modeled in the *cis*-conformation in the density of our structure (Supplementary Fig. 4). This is consistent with Tocilj et al. who remarkably, even though they were unable to visualize this residue in their map, proposed that this proline residue adopts a *cis*-conformation in the full-length structure[12].

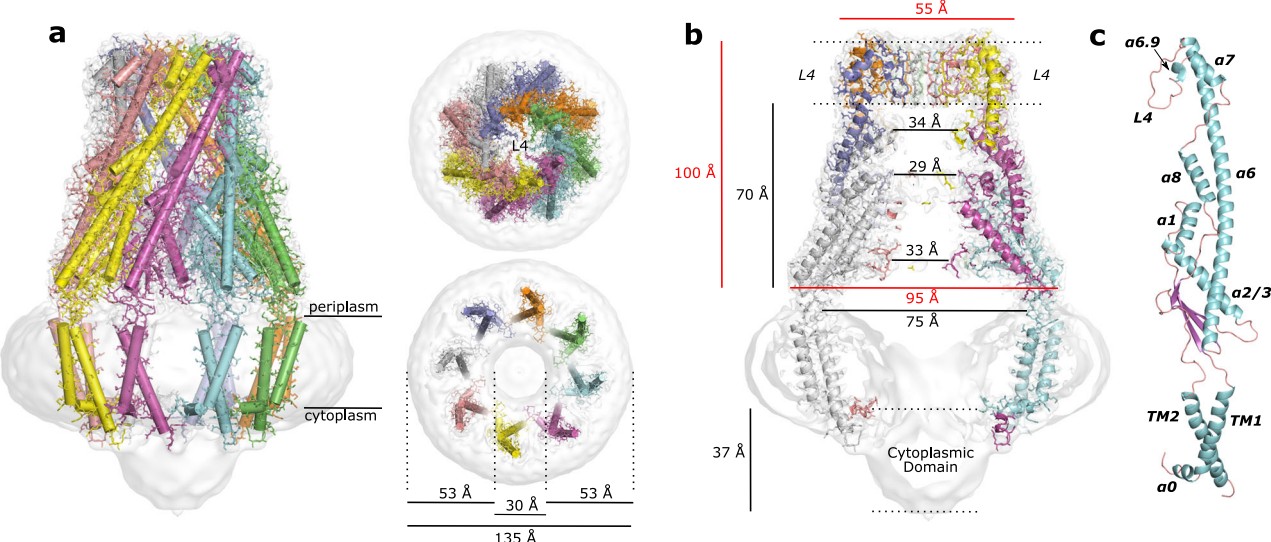

**Fig. 2 The De novo built atomic model of WzzB. a** Cartoon representation of the octameric complex overlaid with the C8-symmetrized WzzB map (white). Top right: top view looking down from the periplasmic side. Bottom right: top view looking down from the periplasmic side, sliced to the level of the membrane to display the doughnut-shaped detergent micelle. **b** Sliced to display the interior of the octameric complex. Red and black values refer to distances measured on exterior and interior surfaces respectively. **c** Single WzzB protomer with labels of the α-helices and loops discussed in the text.

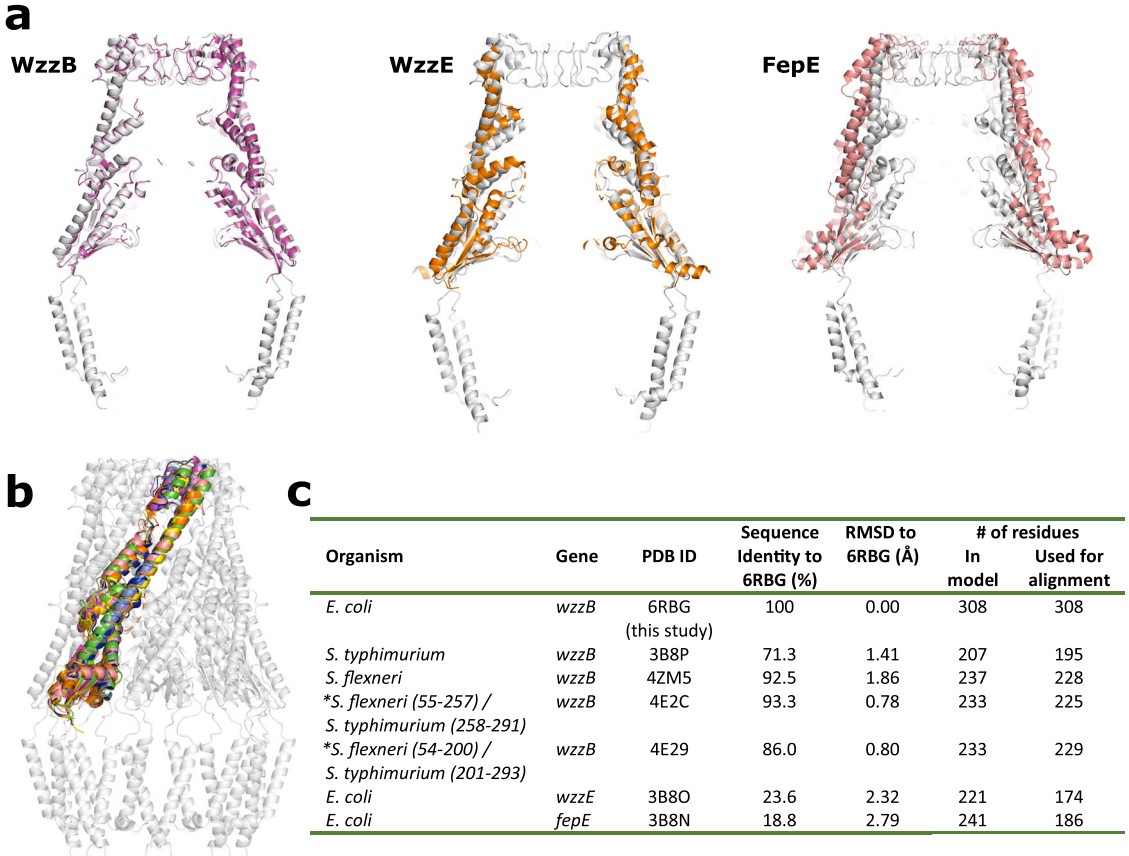

| Organism | Gene | PDB ID | Sequence Identity to 6RBG (%) | RMSD to 6RBG (Å) | # of residues | |
| --- | --- | --- | --- | --- | --- | --- |
| | | | | | In model | Used for alignment |
| *E. coli* | *wzzB* | 6RBG (this study) | 100 | 0.00 | 308 | 308 |
| *S. typhimurium* | *wzzB* | 3B8P | 71.3 | 1.41 | 207 | 195 |
| *S. flexneri* | *wzzB* | 4ZM5 | 92.5 | 1.86 | 237 | 228 |
| *\*S. flexneri (55-257) / S. typhimurium (258-291)* | *wzzB* | 4E2C | 93.3 | 0.78 | 233 | 225 |
| *\*S. flexneri (54-200) / S. typhimurium (201-293)* | *wzzB* | 4E29 | 86.0 | 0.80 | 233 | 229 |
| *E. coli* | *wzzE* | 3B8O | 23.6 | 2.32 | 221 | 174 |
| *E. coli* | *fepE* | 3B8N | 18.8 | 2.79 | 241 | 186 |

**Fig. 3 Comparison of PCP periplasmic domain models to the full-length WzzB. a** Structural alignment of representative periplasmic models that crystalized in the octameric WzzB 4E29 (magenta), WzzE 3B8O (orange), and the nonameric FepE 3B8N (salmon) oligomeric state onto the full-length WzzB 6RBG (gray). **b** Structural alignment of protomers regardless of oligomeric state that they crystallized in onto the full-length WzzB 6RBG. The PDB codes used for alignment are: 3B8P (yellow), 4ZM5 (green), 4E2C (blue), 4E29 (magenta), 3B8O (orange), and 3B8N (salmon). **c** Table comparing the structural alignment of the protomers in (**b**). *Chimeric *S. flexneri/S. typhimurium* WzzB, values in parenthesis are the amino acid residue ranges coming from each WzzB protein[13]. Protomers were aligned and RMSD was calculated using Coot's[44] SSM Superpose function.

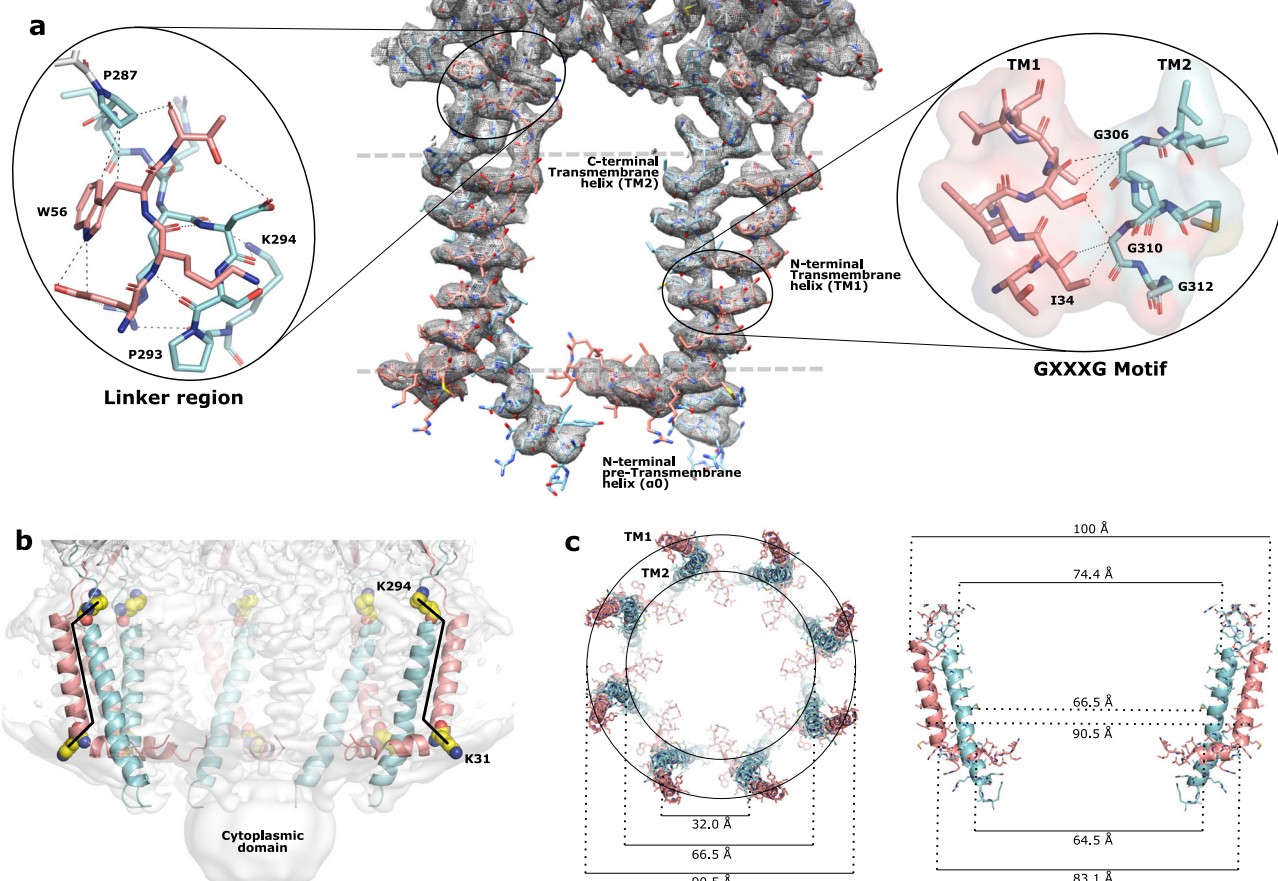

**Fig. 4 The transmembrane domain. a** The C8-symmetrized map and model of the transmembrane domain of two adjacent protomers. Gray dashed lines represent the position of the membrane. Left circle: zoomed area of the linker region. Right circle: zoomed region around the conserved GXXXG motif. **b** The C8-symmetrized map (white) and model showing the transmembrane chamber and cytoplasmic hub. Positions of the conserved K294 and K31 (spheres) are displayed. To visualize the interior of the chamber three protomers have been removed. The black line represents a potential stabilizing brace created by the two lysines on opposite sides of each transmembrane pillar. **c** Dimensions of the transmembrane chamber. Left: view looking down from the periplasmic side sliced to the top of the TM domain. Right: two opposing protomers with distances to measure the diameter of the chamber at different levels within the membrane.

The *cis*-conformation of the ω torsion angle of proline 293 creates a unique kink in this linker region thus helping to direct the C-terminal transmembrane helix into position. As shown through mutational studies[22,23], and consistent with our structure, alteration of this region by removal of this proline residue would dramatically change the three-dimensional structure of this polypeptide stretch and consequently the position of TM2, adversely affecting correct protein folding. Furthermore, analysis using the ConSurf method[27,28] (Supplementary Fig. 5) revealed a conserved lysine (K294) located within this linker region that is likely positioned to interact with the phosphate headgroups to further direct and stabilize the TM domain. This interaction would also likely be disrupted by replacement of the *cis*-P293.

**Transmembrane domain.** High-resolution structural information of the transmembrane domain has to date been elusive. A recent report of a single-particle analysis of the full-length WzzB from *Salmonella enterica*[20] relied on course-grain molecular dynamics simulations fitting into a C12 symmetrized density map in an attempt to analyze the transmembrane domain. However, from visual inspection of the map (EMD-3611), no density corresponding to transmembrane helices could be confidently distinguished above background levels of the detergent micelle. Here, on the other hand, with or without the application of symmetry transformations, the transmembrane helices are clearly

visible above the background of the detergent micelle (Fig. 1), which made it possible to model them with confidence (Figs. 2 and 4). In the octameric assembly, the 2-helix bundles are ~32 Å apart (when measured from center to center) and do not interact with one another. Each of these 2-helix bundles consists of an N-terminal (TM1) and a C-terminal (TM2) transmembrane helix. Within each bundle their closest interactions are at their cross-point, which perhaps not surprisingly, coincides with the conserved GXXXG motif (Fig. 4a) that has been found in ~50% of all α-helical membrane proteins and is suggested to be involved in helix-helix packing[29]. Consistent with that, G306 and G310 are on the inside turn of TM2 at the closest point of contact with TM1 (Fig. 4a). Mutations to WzzB of *Shigella flexneri*[23,30] in this region, except for the double mutation G305A/G311A that conferred very short chain Oags, had little effect on the O-antigen chain-length regulation when these conserved glycines were changed to alanines via single, double, or triple mutations. This is consistent with our structure that shows that there is sufficient space between the two transmembrane helices to accommodate these relatively small structural changes as a result of the mutations. Moreover, residue A41 on TM1 that is directly opposite of G306 is conserved as a phenylalanine in WzzE and FepE, which in the latter two Wzz assemblies would place this bulky side-chain in-between the two transmembrane helices. Similarly, residue S37 on TM1 that is directly opposite G310 of TM2 is not conserved in

other homologs of the family and is often replaced with amino acids having bulky side-chains, such as phenylalanines or methionines. In fact, G310 itself is not strictly conserved and is seen as an alanine in FepE homologs. Thus, glycines (or alanines in the case of FepE) at both of these positions would allow for bulkier residues directly opposite on TM1. The mutational studies[23,30] were performed on WzzB of *Shigella flexneri* that like the corresponding WzzB from *E. coli* contains amino acid residues with small or short side-chains at positions 37 and 41; thus, with the structure presented here, we see now that it is not surprising that these mutations had no effect on O-antigen chain-length regulation.

Each of the transmembrane domains is separated by ~32 Å (measured from center to center) and can be thought of as large vertical pillars that encircle the empty transmembrane chamber with the amphipathic helix (α0) potentially making up part of the chamber floor. Each of the pillars is tilted slightly inward toward the center of the chamber, reducing the inner diameter from ~75 Å on the periplasmic side to ~64 Å on the cytoplasmic side and the outer diameter from ~100 Å on the periplasmic side to ~83 Å on the cytoplasmic side (Fig. 4b, c). As mentioned, each pillar is comprised of the two transmembrane helices with TM2 on the inside and TM1 on the outside (Fig. 4b and c). The inner face of TM2, pointing away from TM1, faces the interior of the chamber and is comprised of exclusively nonpolar, aliphatic residues. TM1 on the other hand is positioned on the outside and analysis using the ConSurf method revealed its outer face to be highly variable (Supplementary Fig. 5) suggesting that it likely does not interact with Wzy or any other potential components of the Wzy-dependent pathway, thus likely playing a structural role in supporting TM2.

Another glycine (G312) of the GXXXG motif is positioned on the opposite face of TM2, pointing away from the neighboring TM1 (Fig. 4a, b) towards the interior of the transmembrane chamber. The double mutation G305A/G311A in *Shigella flexneri* (equivalent to G306/G312 of *E. coli*) conferred only very short chain Oags[23,30] and it was suggested that Wzz protomer interaction occurs via the transmembrane domain, and this mutation results in the failure of the protomers to interact[30]. However, our structure demonstrates that each transmembrane domain of the WzzB protomers are 32 Å apart with no possibility of interaction at or near the GXXXG motif. Since our structure is lacking the polymerase Wzy and G312 is on the opposite face of TM2 pointing away from TM1, into the chamber, it stands to reason that a more likely explanation is that in a fully intact Wzy-dependent supercomplex Wzy would be inserted either partially or fully into the chamber for one of its transmembrane helices to be positioned close enough such that this double mutation would disrupt Wzz:Wzy interactions resulting in the loss of chain-length regulation and the production of unregulated Oags.

On the cytoplasmic side of the membrane, TM2 continues straight through the predicted membrane region until the C-terminal residue. However, at the N-terminal end of the protein, a short amphipathic helix (α0) runs parallel to the membrane and interacts with TM2 of its helix bundle as well as with TM2 of the neighboring protomer, which curves slightly toward α0 (Fig. 4 and Supplementary Fig. 6). The highly conserved K31 is positioned at the start of the TM1 that creates an elbow between TM1 and α0 causing a sharp 90° turn at the cytoplasmic membrane interface, resulting in α0 pointing towards the center of the octameric complex. Consistent with modeling and atomistic simulations[20] that suggest K31 to be localized within the phosphate band and interacting with the phosphate groups of the lipids, K31 in our structure is located at the outer edge of the detergent micelle. Mutations to K31 result in an inactive complex[23] and Collins et al.[20] suggest that mutations to this residue cause TM1 to be

pulled deeper into the membrane, thus disrupting TM-TM interactions. That is also possible; however, since K31 is at the elbow between α0 and TM1, another explanation based on our structure is that the position of α0 is dramatically altered, disrupting its interactions with the neighboring TM2 and likely the overall protein stability. Lysine at this position is likely interacting strongly with the phosphate groups of the lipids creating a rigid structure that helps to maintain α0 in its correct position allowing it to interact with neighboring TM2s and potentially bringing added stability to the octameric complex.

**Cytoplasmic domain.** The cytoplasmic domain is considerably lower in resolution and can be seen as a hub positioned at the center of the complex at the cytoplasmic membrane interface and extending downward away from the membrane interface (Fig. 4b). This cytoplasmic domain (not modeled due to low resolution in this area) is where the first ~15 amino acids of the N-terminus and the C-terminal 8×His tag used for affinity purification from each protomer are located. Similar to the central hub of a wheel, like spokes, the 8 α0 s from each protomer can be seen radiating outward at the predicted membrane interface of this domain. In addition, the C-terminus of TM2, containing the non-cleavable 8×His tag used for purification, can be seen connected to the cytoplasmic hub at lower threshold levels (Fig. 4c, Supplementary Fig. 6). As mentioned above, and in line with previous findings[12,19]; we also found WzzB particles as dimers of the octameric complex in an apparent pseudo D-symmetry that associate via this cytoplasmic domain into dimers. Attempts to refine these dimers of octamers resulted in a low-resolution structure that two WzzB octamers could be docked into. Further masking and refinement of a single octamer within a dimer resulted in a 3.9 Å resolution structure where each transmembrane pillar and α0 are in a similar conformation as the monomeric form of the octamer suggesting that this dimeric association does not affect the conformation of the transmembrane domain.

In an attempt to better characterize the cytoplasmic domain and understand the role of the cytoplasmic N-terminal residues, a truncated WzzB, missing residues 1–15 was created. Purification of this truncated WzzB resulted in a similar size-exclusion chromatograph profile as seen for the full-length protein, and single-particle analysis of population 1 revealed similar particles as observed for the full-length protein (Fig. 5, Supplementary Fig. 7). However, unlike the full-length protein, virtually all of the particles of the N-terminally truncated variant are of a dimeric form. Thus, as done for the full-length protein, masking and refinement around half of the dimer resulted in a 4.5 Å resolution octameric structure with, remarkably, its transmembrane pillars and α0 in similar positions and conformation as observed in the full-length protein suggesting that these N-terminal residues do not contribute to the stabilization of the transmembrane domain. This is perhaps not surprising due to the disordered nature of these residues in the full-length octamer. In addition, both the C1 and C8 reconstructions of the N-terminally truncated WzzB (Supplementary Fig. 7) are comparable to the resolution obtained of the similarly processed full-length reconstructions when masking around and refinement of a single octamer within the dimeric form (Supplementary Fig. 8) was performed suggesting that removal of the N-terminal residues did not significantly affect the resolution obtained. With residues 1–15 removed, the cytoplasmic domain of the truncated WzzB consists solely of the 8×His tag used for purification. It noticeably smaller in diameter and is centered lower down from the membrane interface when compared to the full-length WzzB (Fig. 5c). Given the resolution limits of this region of the protein, it is difficult to judge what, if

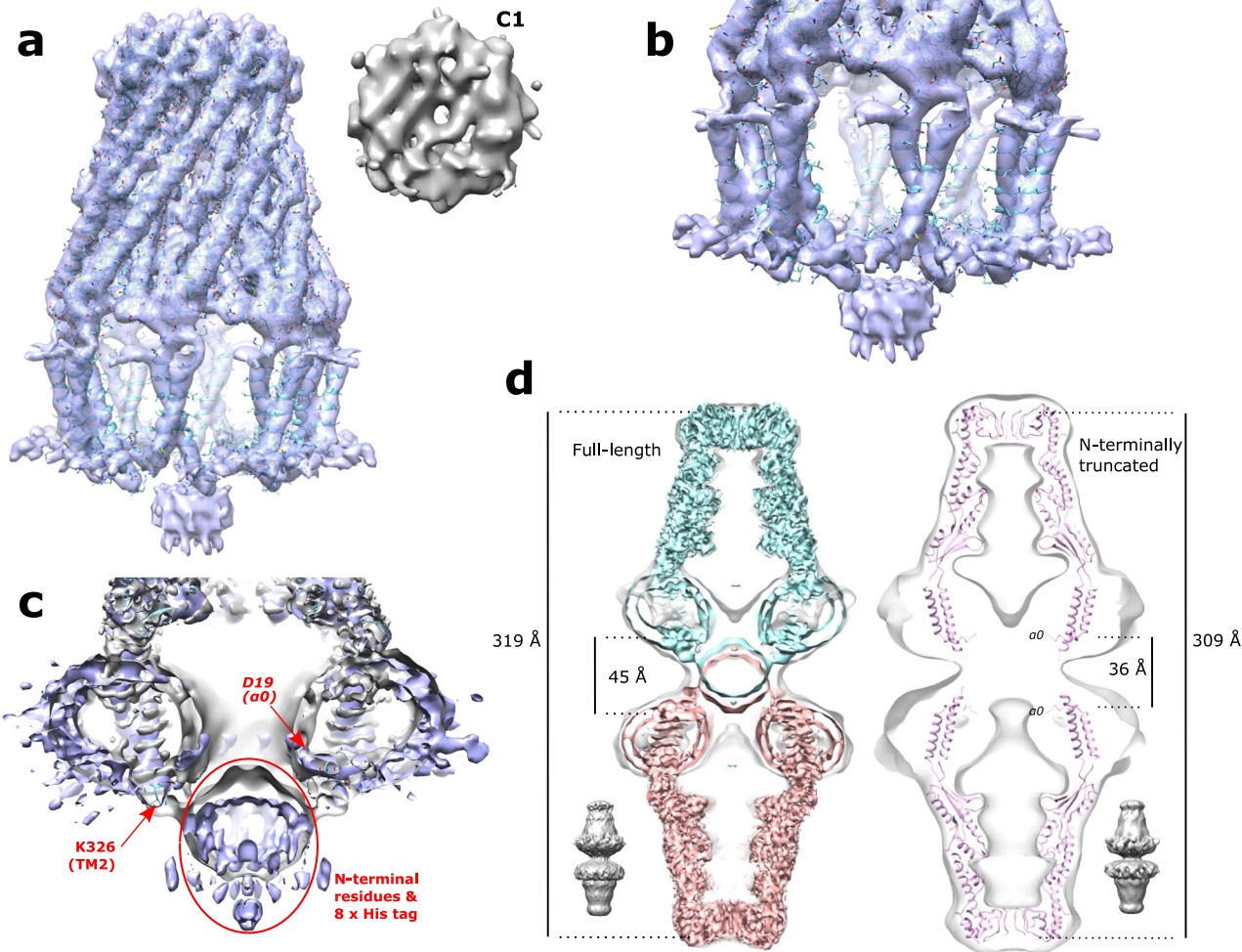

**Fig. 5 Structure of N-terminally truncated WzzB. a** Docked atomic model to the N-terminally truncated WzzB C8-symmetrized map. Top right: view from the top of the periplasmic domain showing an octameric complex similar to the full-length WzzB in the unsymmetrized map. **b** Zoom of the transmembrane region, showing a similar architecture to the full-length WzzB. **c** Overlay of the full-length (gray) and N-terminally truncated (purple) WzzB. An equivalent threshold was chosen based on the visually similar size of the detergent micelles of both maps. **d** Docking into the dimeric forms. Left: to show the overlap of the two cytoplasmic domains in the dimeric form, two full-length monomeric octamer maps (cyan and salmon-colored) are docked into the full-length dimer (gray outline and bottom left). Equivalent thresholds were chosen based on the visually similar size of the detergent micelles of the monomeric and dimeric forms. Right: two atomic models docked into the N-truncated dimer (gray outline and bottom right).

any, is the influence of the expression tag on the position of the N-terminal residues. However, based on the comparison of the two structures it seems likely that the bulk of the N-terminal residues are positioned between the 8×His tag and the membrane interface, perhaps contributing to the floor of the transmembrane chamber.

Refinements of dimers of the full-length and truncated protein resulted in similar low-resolution reconstructions, however, the dimer of the full-length protein was noticeably 10 Å longer at 319 Å compared to the truncated version at 309 Å in length (Fig. 5d). Further docking of two atomic models of the monomeric octamer into the low-resolution dimers of both the full-length and the truncated WzzB revealed a distance between the two octamers of 45 Å in the full-length dimer and 36 Å in truncated dimers when measured from α0 to α0 (Fig. 5d) consistent with the removal of the N-terminal residues. Furthermore, docking of two refined maps of the full-length monomeric octamer into its corresponding dimer resulted in a significant overlapping of the 2 cytoplasmic domains suggesting at least a partial intertwining of these domains in the dimeric form, likely the C-terminal 8×His tag used for purification.

## Discussion

**Stabilization of the transmembrane domain**. The overall structure presented here is consistent with previous crystal structures of the periplasmic domain from various PCP family members, and provides valuable information on the transmembrane region of this family of proteins. The fact that we are able to resolve to high resolution the eight transmembrane pillars that are separated by 32 Å in the cryo-EM map without any additional components bound to stabilize them speaks to their very rigid nature. Except for the disordered cytoplasmic N-terminal residues, whose removal had little effect on the positioning of the transmembrane domain and overall architecture of the complex (Fig. 5), it is likely that the fold of the protomer and the overall nature of the circular, octameric WzzB complex help create a very stable structure that permitted its high-resolution structure determination. For example, similar to the side-by-side packing of helices seen in the periplasmic domain, side-by-side packing of the amphipathic α0 with adjacent TM2s and the crossing of TM1 and TM2 at the GXXXG motif within each of the transmembrane pillars will likely have stabilizing effects. In addition, on the periplasmic side of the membrane, the conserved

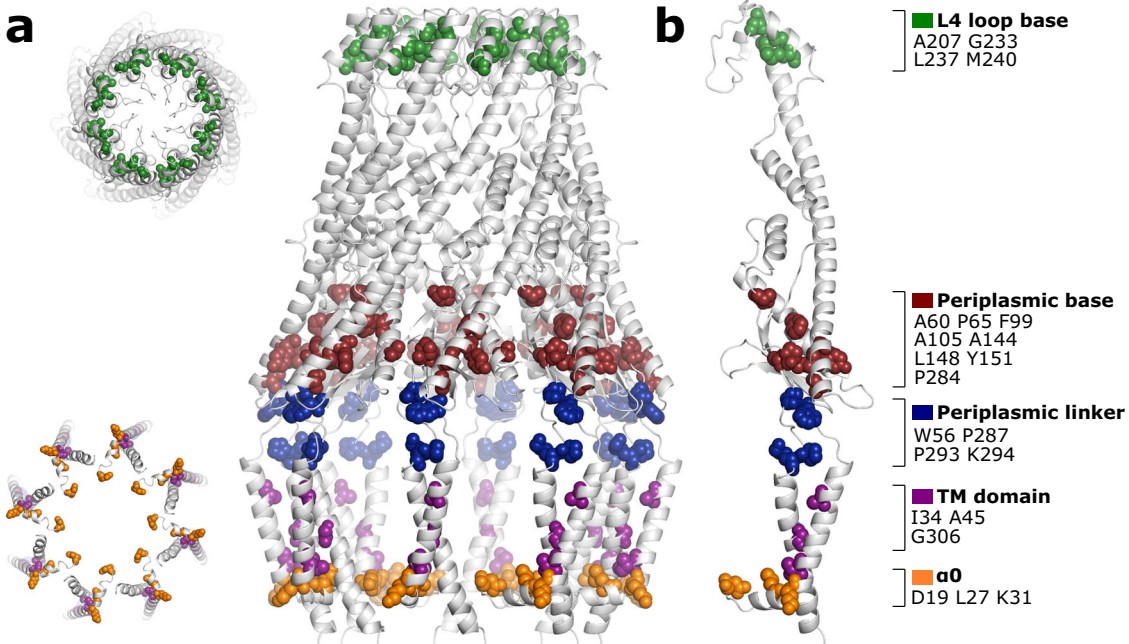

**Fig. 6 Location of conserved residues mapped to WzzB.** The conserved residues (spheres) can be grouped into five distinct regions: The L4 loop base (green), periplasmic base (maroon), periplasmic linker (blue), TM domain (magenta), and α0 (orange). **a** The octameric WzzB complex. Top left: view looking down from the top of the periplasmic domain. Bottom left: view looking up from the cytoplasm. **b** A single protomer of WzzB. Conserved residues were determined using the ConSurf method[27,28].

*cis*-conformation of P293 and interactions within the linker region are likely creating local mini-cores (Fig. 4a, circled left) around each transmembrane pillar that add additional stability to the pillars.

Our structure suggests that the conserved K31 is positioned to contribute to multiple braces that play a central role in the stabilization of the transmembrane domain. With TM1 positioned on the outside of TM2, K31 creates an elbow directing the amphipathic α0 toward the interior of the transmembrane chamber, crossing from the outside to the inside acting like a brace to TM2, adding additional stability to the pillar. This directing of α0 allows this amphipathic helix to interact with adjacent TM2s creating additional stabilization (Supplementary Fig. 6). Further stabilizing effects are likely created by the positively charged K294 and K31 at the periplasmic and cytoplasmic side of the membrane, respectively, that are likely interacting with the negatively charged phospholipid headgroups. They are not only on the opposite membrane interfaces but their side chains are positioned to point to opposing sides of the membrane chamber; K294 toward the interior and K31 to the exterior potentially creating a membrane-spanning brace that further stabilizes the membrane pillars (Fig. 4b). The central role of K31 in transmembrane stability is consistent with previous studies that have shown that mutations to this single residue had an adverse effect on protein folding and expression levels of WzzB[23].

Analysis of the *E. coli* WzzB using the ConSurf method revealed that conserved residues of the class 1 PCPs could be grouped into five distinct regions (Fig. 6): L4 loop base, periplasmic base, periplasmic linker, TM domain, and α0. As mentioned above, various mutational studies to residues within these regions largely result in complexes of low activity resulting in unregulated Oag chain-lengths being detected, under-pinning the importance of these regions. With the exception of the L4 loop, these regions likely play mainly a structural role in correct protein folding, interaction with the polymerase Wzy and the

UndPP-linked O-Ag, and stabilizing the Wzy-dependent super-complex. Since many of the residues and motifs that have been suggested to stabilize the TM domain are conserved across the PCP family of proteins (eg. K31, P293, K294, GXXXG motif), the question arises as to how we were able to resolve the transmembrane domain without any additional stabilizing components in contrast to other attempts that could not[19,20]. The fact that the majority of the WzzB that was extracted from the membrane lacks a coherent structure (Supplementary Fig. 1) could suggest a highly instable or dynamic complex. Perhaps, simply the fact that our purified WzzB was concentrated and frozen onto cryo-EM grids immediately (or within a maximum of a few hours) after purification allowed us to resolve the transmembrane domain in a significant fraction of the sample.

**Insights into the architecture and mechanism of the Wzy:Wzz complex**. Although the molecular mechanism of LPS assembly is currently not known, as summarized and critiqued by Islam and Lam[10], many intriguing models have been postulated over the years. For example, it has been suggested that Wzz acts as a molecular timer, allowing polymerization to occur for only a pre-set amount of time[31], before ligation to the core-Lipid A. An alternative model suggests that Wzz is a molecular chaperone mediating the interaction between the polymerase Wzy and the ligase WaaL where the stoichiometry between the two determines the length of the polysaccharide[23,32]; while yet another model suggests that Wzz interacts with both Wzx and Wzy[33]. An opposing model proposes that Wzz acts as a central hub with Wzy encircling it and that chain elongation is occurring by the transfer from one Wzy molecule to the next[12]. A recent proposal based on a low-resolution cryo-EM structure combines aspects of the molecular ruler and timer mechanisms[20]. Although this structure provides valuable information on the transmembrane region of this family of proteins it is still difficult to give a detailed critique to validate or exclude any of the above proposed

mechanisms beyond what Islam and Lam have already discussed[10] without the structure of a fully intact Wzy-dependent supercomplex.

However, this structure does support the idea that the Wzy polymerase and Wzz co-polymerase must work in close contact in order to properly modulate the growing polysaccharide chain length[11,34], and thus would likely constitute the complex at its minimum. The 32-Å gap between the TM helix pillars of each protomer, the large empty transmembrane chamber, and the location of residues within the transmembrane domain support this idea. Although there is currently no 3D structural information about Wzy proteins, based on helix packing of other large membrane protein complexes[35], we hypothesize that an object of what would likely be the size of Wzy could possibly slot into the gap between each transmembrane domain with the large empty chamber providing ample space for a Wzy molecule. Either a partial or complete insertion of a Wzy molecule into the chamber of Wzz would create multiple contact points between the partners. For example, insertion of Wzy would likely create contact points with G307 and G312 of WzzB that are located on the inner face of TM2 pointing into the interior of the empty chamber. This is consistent with a double-mutant that includes the equivalent G312A that led to the formation of unregulated Oag chains[30] and also in agreement with a report on the formation of a stable Wzy:Wzz complex, which suggests that many regions of Wzy are interacting with Wzz[34].

Previous studies have revealed that multiple, mainly surface regions on both the interior and exterior of the periplasmic bell determine the modal length of the growing polysaccharide[13,22,23,26,36,37]. Similarly, an electrostatic surface potential analysis of WzzB (Supplementary Fig. 9a) also revealed negative patches on both the outer and inner surface of the bell that could also suggest that both interior and exterior surfaces could interact with a growing Oag. If the growing Oag is on the outer surface it could be possible that multiple polysaccharide chains can grow along Wzz at any given time (Supplementary Fig. 9b). For any one of these growing polysaccharides there would be two or three major degrees of freedom between each pair of sugar residues. Such a large number of degrees of freedom in the polysaccharide could lead to similar head-to-tail distances, albeit with a small difference in the number of repeating units (RUs). This high degree of freedom could also result in multiple paths up the outer surface of the Wzz scaffold as the polysaccharide grows. Alternatively, the single point mutation A107P[14] located on the inner face, at the base of α2/3 resulted in a shortened Oag chain-length distribution. This study also revealed a cooperative binding of very-short Oags with a reduced affinity of the A107P mutation that could suggest that multiple short UndPP-linked Oags could bind inside the bell simultaneously. Consistent with that, WzzB is widest at the periplasmic interface and could accommodate multiple short UndPP-linked Oags at a time. However, due to a narrowing of the periplasmic bell (Fig. 2), steric hindrance would likely limit one growing Oag inside the bell at any given time (Supplementary Fig. 9b). Growing inside the bell would also likely limit the conformational freedom of the growing polysaccharide.

Blind docking of an oligosaccharide containing 2 repeating units (2-RU), representing the terminal part of the *E. coli* O16 O-antigen polysaccharide, to the top portion of the periplasmic domain (Supplementary Fig. 9c) suggests that it is possible for a single polysaccharide to be inserted into the pore created by the 8 L4 loops. This would suggest that once a growing polysaccharide reaches the top of the periplasmic domain, from either the inside or outside, it is captured by the L4 loops to somehow stabilize it to allow for its continued elongation. This is consistent with previous studies that the L4 loop is required for the synthesis of long polysaccharide chains[13,22,26]. Thus, taken together we

speculate that Wzz could be acting as a central hub where short UndPP-linked Oags could pool to be used in the elongation process as well as a scaffold for the growing Oag. This is where one or more Wzy molecules could associate and dissociate at a time with Wzz and its pool of UndPP-linked Oags via the 32-Å gap to carry out the polymerase reaction. Regardless of whether the growing polysaccharide is elongated on the inner or outer surface of the bell, a single polysaccharide at a time is likely captured by the L4 loops to somehow stabilize its association with Wzz to allow its continued growth. It is conceivable that the growing polysaccharide could even pass through or be exported through the hydrophilic, glutamine-rich L4-pore to possibly continue growing on the inner or outer surface of Wzz, potentially explaining the multiple, seemingly contradictory studies that both the inner and outer surfaces of Wzz determine the modal length of the growing polysaccharide[13,22,23,26,36,37].

Overall, this study reveals structural details of the transmembrane and cytoplasmic domains as well as the highly conserved proline-rich segment proximal to the C-terminal transmembrane helix of the polysaccharide co-polymerase family of membrane proteins. The structure and hypotheses presented here provide a solid base for future studies aimed at elucidating the structure and mechanism of a fully intact Wzy-dependent supercomplex.

## Methods

**Protein expression and purification**. The full-length WzzB gene was amplified from *E. coli* K12 genomic DNA using standard PCR techniques using the primers EcWzzB_fwd and EcWzzB_rev (Table S1). Similarly, an N-terminal truncated WzzB, missing the first 15 amino acids was created by the introduction of a start codon at position 16 creating a truncated E16M variant. This variant was created using the primers EcWzzB_15_fwd and EcWzzB_rev (Table S1). Both the native and the truncated genes were inserted into a modified pWaldo vector[38] that replaces the GFP reporter with a C-terminal 8×His tag[39] using the restriction sites *XhoI* and *EcoRI* and their sequences confirmed by DNA sequencing. For large-scale protein purifications, plasmid expressing WzzB was transformed into *E. coli* C41. A pre-culture from a single colony was grown overnight at 37 °C in LB broth containing 50 μg/mL kanamycin, which was then used to inoculate 1.5 L LB broths in a LEX expression system. Cells were grown at 37 °C until an $OD_{600} = 1.0$ prior to the induction of protein expression with 0.4 mM IPTG at ambient temperature for 4 h. Cells were then harvested via centrifugation at $6000 \times g$ for 15 min and frozen at −20 °C. Frozen cells were thawed and suspended in lysis buffer (50 mM TRIS pH 8.0), and lysed by passage of at least three times through an Emulsifex cell disrupter (Avestin). The lysate was centrifuged at $16,000 \times g$ for 40 min to pellet cell debris. Membranes were then isolated by ultra-centrifugation of the cleared lysate at $40,000 \times g$ for 45 min; washed one time with a washing buffer (500 mM NaCl, 50 mM TRIS pH 8.0); ultra-centrifuged again, and finally suspended in buffer containing 300 mM sucrose, 20 mM TRIS, pH 8.0, and frozen at −80 °C.

Purified membranes were diluted tenfold with buffer A (50 mM TRIS pH 8.0, 400 mM NaCl, 5% glycerol, 10 mM imidazole, 5 mM β-mercaptoethanol) and solubilized with 1% DDM at 4 °C for 90 min, followed by centrifugation at $40,000 \times g$ for 45 min. The supernatant was incubated on a mixer with Ni-NTA resin (Qiagen) equilibrated with buffer A supplemented with 0.05% DDM for 90 min at 4 °C. The resin was collected with a gravity flow column and washed with 100 mL buffer B (50 mM TRIS pH 8.0, 200 mM NaCl, 5% glycerol, 20 mM imidazole, 5 mM β-mercaptoethanol, 0.04% DDM), followed by a second 100 mL wash with buffer C (50 mM TRIS pH 8.0, 200 mM NaCl, 5% glycerol, 40 mM imidazole, 5 mM β-mercaptoethanol, 0.04% DDM). The protein was eluted from the Ni-NTA resin with buffer D (50 mM TRIS pH 8.0, 150 mM NaCl, 5% glycerol, 250 mM imidazole, 5 mM β-mercaptoethanol, 0.02% DDM). The eluted protein was concentrated with a 100 MWCO ultrafiltration spin column (Vivaspin) to <5 mL and further purified by injection into the HiLoad 16/60 Superdex 200 column (GE Healthcare) equilibrated with buffer E (25 mM TRIS pH 8.0, 100 mM NaCl, 3% glycerol, 0.5 mM TCEP, 0.02% DDM) at 4 °C. The appropriate fractions were pooled and the final purified protein was concentrated with a 100 MWCO filter to 3 mg/mL, and snap frozen in liquid nitrogen, and stored at −80 °C if not used immediately.

**Gel electrophoresis**. Nu-PAGE Bis–Tris gradient (4–12%) gels (Thermo Fisher Scientific) were loaded with the appropriate samples mixed with loading buffer (40 mM Tris-HCl, pH 6.8, 8% glycerol, 1% SDS, and 1 mg/mL bromophenol blue) and run at 200 V for 40 min in MES running buffer. The gels were stained with a standard 3 mg/mL Coomassie R-250 staining solution and destained with a 10% ethanol, 10% acetic acid solution. Novex Native gels (4–16%) Bis–Tris (Thermo Fisher Scientific) were loaded with the samples. Light-blue or clear buffer was used as the cathode buffer. The gels were run at 150 V for 2 h 30 min at 4 °C.

NativeMark Unstained Protein Standard (Thermo Fisher Scientific) was used as a marker. The gels were stained and fixed according to the manufacturer's protocol.

**Negative-stain electron microscopy**. Three microliters of appropriately diluted, purified WzzB was applied to a carbon-coated EM grid previously glow-discharged under vacuum for 1 min at 15 mA (PELCO easiGlow), incubated for 1 min, then blotted to remove excess liquid. The sample was stained with 5 μL of 0.5% uranyl acetate, incubated for 1 min, then blotted to remove excess liquid. Digital electron micrographs were recorded with a FEI Tecnai G2 Spirit electron microscope operating at 80 kV equipped with a Gatan US 1000 P CCD camera. Images were collected at a magnification of ×49,000, corresponding to a pixel size of 2.1 Å.

**Cryo-EM sample preparation and data collection**. Purified protein at 3.0 mg/mL (3 μL) was applied to cryo-EM grids (C-flat 2/2-3C-T-50) previously glow-discharged under vacuum for 40 s at 20 mA (PELCO easiGlow), incubated for 30 s, blotted for 3 s, and plunge frozen in liquid ethane using a Vitrobot Mark4 grid freezing device (FEI) with the chamber maintained at 4 °C and 100% relative humidity. An optimized grid was imaged with a Thermo Scientific Titan Krios G3 electron microscope equipped with a K2 camera, operating at 300 kV. 2347 movies of 40 frames each were acquired in electron counting mode at 1.06 Å/pixel, 1.30 electrons/pixel/s, with a total exposure time of 60 s and stacked into a single MRC stack using EPU automatic data collection control software with defocus values ranging from 1.4 to 3.2 μm. Purified N-terminal truncated protein was prepared for cryo-EM imaging as described for the native protein. An optimized grid was imaged with a Thermo Scientific Titan Krios G3 electron microscope equipped with a Falcon III camera, operating at 300 kV. 2050 movies of 40 frames each were acquired in electron counting mode at 0.85 Å/pixel, 1.25 electrons/pixel/s, with a total exposure time of 60 s and stacked into a single MRC stack using EPU automatic data collection control software with defocus values ranging from 0.8 to 3.0 μm. For data collected with the Falcon III camera, the final movie frame was removed during image processing due to the appearance of the camera shutter.

**Cryo-EM image processing**. Contrast transfer function (CTF) parameters were estimated from averaged movies using CTFFIND4[40] and initial particle images were selected manually and subjected to 2D classification in cryoSPARC v2[41]. Automatic particle selection was performed with templates from the initial 2D classification. Beam induced particle motion between fractions was corrected with an implementation of alignparts_lmbfgs[42] in cryoSPARC v2. The number of particle images were reduced by further 2D and 3D classifications and hetero-genesis refinements (Supplementary Fig. 3). Initial maps were calculated ab initio and the final map was refined with cryoSPARC's non-uniform refinement feature[43] with and without the application of C8 symmetry. Both the full-length and the N-terminally truncated WzzB were processed similarly (Supplementary Fig. 7). Maps of the full-length and N-terminally truncated WzzB have been deposited in the Electron Microscopy Data Bank under accession code EMD-4791, EMD-4798, EMD-11908, and EMD-11909, respectively.

**Model building and refinement**. A final 3.0 Å resolution, C8-symmetrized density map was used for the de novo model building of the asymmetric unit of the full-length WzzB complex. Manual building in Coot[44] was performed starting with easily interpretable features from the density map, such as bulky residues and alpha-helices. After anchor points were established, a near-full model was then able to be built with the aid of secondary-structure predictions from Jpred[45] and Protter[46]. Once completed, the asymmetric model was further improved through iterative rounds of refinement with phenix_real_space_refine[47] and fixing with Coot. After the asymmetric unit was completed in this way, C8 symmetry was applied to the model with the Phenix suite of programs[47]. This full complex was then refined again with phenix_real_space_refine, and further improved through iterative rounds of refinement with phenix_real_space_refine and fixing with Coot. The final model was validated with MolProbity[48] and EMringer[49] and has been deposited in the Protein Data Bank under the PDB accession code 6RBG. All figures were made with UCSF Chimera[50] or PyMOL[51]. The polysaccharide was built using LPS Modeler[52] and then manually docked using UCSF Chimera. Blind docking was performed with SwissDock[53,54].

## Data availability
The cryo-EM density maps and atomic coordinates have been deposited in the Electron Microscopy Data Bank and Protein Data Bank under the accession codes EMD-11908, EMD-11909, EMD-4791, EMD-4798, and 6RBG, respectively. All relevant data supporting the key findings of this study are available within the article and its Supplementary Information files or from the corresponding authors upon reasonable request. Additional maps and source data are available from the corresponding authors upon request.

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

## Acknowledgements

This research was funded by the Swedish research council (2017-04018, 2017-03703) and the Knut and Alice Wallenberg Foundation (2017.0275). Negative-stain electron microscopy was carried out at the Imaging Facility at Stockholm University (IFSU). We would like to thank Agnes Moe for the technical assistance in acquiring the negatively stained images. Cryo-EM sample screening, optimization, and data collection were performed at the Cryo-EM Swedish National Facility in Stockholm, Sweden funded by the Knut and Alice Wallenberg, Family Erling Persson and Kempe Foundations, SciLifeLab, Stockholm University, and Umeå University. We would like to thank Marta Carroni for the technical assistance in Cryo-EM data collection.

## Author contributions

B.W. and M.H. conceived and designed the research. B.W. collected the final cryo-EM data, performed image analysis, various calculations with the cryo-EM data, and built the de novo atomic model. B.W. performed gene cloning, protein purification, and negative-stain EM analyses. B.W. and R.G.N. performed SEC, Blue-native, and SDS-PAGE analyses. B.W. and R.G.N. performed cryo-EM sample preparation, and the initial screening, data collection, and analysis. B.W. and G.W. performed LPS modeling and docking. B.W., R.G.N., G.W., and M.H. interpreted the data. B.W., G.W., and M.H. wrote the manuscript. B.W. prepared data figures and legends.

## Funding

## Competing interests

The authors declare no competing interests.
