## [Peer Review File · Nature Communications]

REVIEWER COMMENTS

Reviewer #1 (Remarks to the Author):

Several high resolution structures of the periplasmic segments of the bacterial polysaccharide co-polymerases, components of the O-antigen biosynthesis apparatus that are crucial in determining the length of the synthesized polysaccharide, have been determined over the last dozen years. However, the previous attempts by several labs to determine the structure of the full-length co-polymerase oligomer resulted in only low resolution structures, in which the transmembrane segments were not well resolved. Wieseman and colleagues succeeded in this endeavour and present in this manuscript the near-atomic resolution structure of the full-length E. coli polysaccharide co-polymerase determined by cryo-EM. Visualization of the transmembrane domains identified in addition to the two predicted TMs also a short N-terminal helix $\alpha 0$ aligned along the membrane surface that was not anticipated from the sequence. Together, this structure adds substantially to our understanding of co-polymerases and provides better starting point for hypothesis of how this structure controls the polymer length during biosynthesis.

While the structure shows for the first time the structural details of the transmembrane domains, the subsequent analysis of the structure could/should be much more thorough. The likely reasons why previous crystallographic and cryo-EM studies did not resolve the TMs, while the periplasmic domains were relatively well defined, is the flexibility of the linker regions between the TMs and the periplasmic segments and/or the effect of the detergent/solvent destabilizing the linkers. The helix bundles from different protomers are widely apart yet the authors do not speculate on what keeps them in the same relative orientation in every protomer. Are there small minicores within the linkers that provide some rigidity to the linkers? In what way Lys31 and Lys294 stabilize the two-helical bundles? What is the role of the N-terminal ~ 15 residues? Are they essential for the structural integrity of the oligomer? From the figures, it appears that they all come together in the center but are disordered. Would the construct with deletion of 15 N-terminal residues give similar cryo-EM particles? This would be a relatively easy experiment to perform.

There is a chapter on the cytoplasmic domain but since it was not modeled in the structure, there is really nothing to say about it in the result section. By the way, which residues were included in the model? The N-terminal residues shown in Figure 4 do not fit well the presented density.

Figure 1 leaves me with a question about modeling loop L4. In the C1 model there is a hole in the center of the oligomer indicating low density in this region. Where then the density in this region arrives from in the C8 model? Averaging weak density should still result in weak density.

In Figures 2d and S2a I can see many double octamers, by my eye there more than 25% of them. Is there anything about the TM region from this reconstruction that show a similar arrangement of the TM helices? Does the thinner connection between the two octamers correspond to the cytosolic region or is it still part of the TM region? Superposition of the high-resolution octamer model on the low-resolution double octamer would be very useful and, hopefully, informative.

Discussion section should be reconsidered. It presents several hypotheses proposed from the structure of the model but there is no evidence at all presented in their support. The first chapter on a potential use of the periplasmic oligomer as a scaffold for creating larger assemblies of smaller transmembrane proteins for application in cryo-EM is interesting but has nothing to do with the result section and certainly should not be mentioned at the beginning of discussion, if at all. If there is data showing that this is indeed useful then it would be great.

I do not see any new evidence from the structure reflecting on the mechanism of polysaccharide oligomers assembly and length control. The authors list models proposed in the literature but provide no data in support of any particular model. The discussion of a potential interaction between transmembrane segments of WzzB and Wzy polymerase is a bit of a hand waving with no data to support this. Many labs attempted to show physical interactions between WzzB and Wzy but although this is most likely, there is no direct evidence as yet of this association. It is possible that there might be 8 copies of Wzy around WzzB but the hypothesis of simultaneous synthesis of 8 polysaccharide chains around WzzB needs some evidence. WzzB presence leads to synthesis of polysaccharide chains that contain well over 20 repeating units of 4 sugars and the length of this polymer would exceed

significantly the 100 Å distance from the bottom to the top of the periplasmic domain. So even if the tip of the polymer is locked by the L4 loop, what is the signal to stop adding new repeating units? In summary, I am really excited about the new structural details of the transmembrane segment of the co-polymerase WzzB but at the same time I would like to see a more in-depth analysis of the structure, in particular what keeps the TM aligned the same way in all protomers. Also, at least some data/evidence in support of the various hypotheses presented in the discussion.

Reviewer #2 (Remarks to the Author):

The authors describe the first full length structure of the O antigen (Oag) polysaccharide co-polymerase, WzzB of E. coli K12. Of particular significance is that they were able to determine at high resolution the structures of the TMs regions (TM1 and TM2), the tip region at top of the periplasmic region/bell, and identify a short alpha helix ($\alpha 0$) at the amino terminal end of the protein. These were not visible in other structural studies. This new insight greatly helps to explain the impact of many mutagenesis studies targeting conserved residues that were previously published. The authors do not provide any new mutagenesis/LPS phenotype data based on their structure but interpret existing mutagenesis and cryoEM data in the context of their structure. While much of this is reasonable, aspects related to interaction of WzzB and with the Oag polymerase WzyB and Oag itself are highly speculative, and somewhat inconsistent with published data, and even their own WzzB structure. In several places in the Results and Discussion the experimental facts are misinterpreted and oversimplified.

The authors need to better define their terms such as less expression. In the absence of WzzB, LPS with Oag is still produced but it lacks modal length control and hence has an unregulated chain length. WzyB is working quite well without WzzB.

The authors speculate and model the location of the growing Oag polymer, and wrap it around the outside of the WzzB octamer. Is this energetically even feasible?

WzzB protein act on a diverse wide range of Oags (and a wide range of WzyB proteins). While they authors have used the O16 (as their WzzB comes from K12), this modelling needs to be done with different Oags to take into account this low specificity.

The authors mention the A107P mutant completely out of context when discussing TM interactions (p. 8. I. 8-11). What they fail to consider is the Chang et al determined that the A107P mutant has weaker binding to Oag, and that the A107 is located on the inner side of the WzzB dome. Furthermore, most mutations (in WzzB, FepE) affecting Oag chain length also map to the inner side of the dome. Their discussion and model ignores this data.

The authors have used a WzzB protein with a Hisx8 at the amino terminal end. Do they know if the protein is functional and confers a WT LPS phenotype? How does this poly His extension impact their interpretation of the base region / ($\alpha 0$) in the cytoplasm?

The authors are unable to explain why the vast majority of the extracted WzzB is in a form that lacks a coherent structure.

The authors study while consistent with other studies (i.e. octameric) fails to acknowledge that in all cases the most stable form of the protein is that which is used for structural studies. In vivo cross linking detects multiple forms of WzzB, so the evidence for the stable WzzB scaffold structure that authors model is weak. Indeed, the Wzz:WzyB interaction is likely to be weak based on published attempts to detect it. It is more likely that the WzzB self-interaction is dynamic in nature and hence structure.

The authors' excellent description of the orientation and function of conserved Gly residues in TM2 is inconsistent with their overall model for interactions between Wzz and Wzy. G312 is located inside the bell and is proposed to interact with Wzy. While taking this into account, their model places Wzy largely outside the bell. Would it not be simpler if Wzy is mostly inside the bell and polymerises an Oag that grows inside the bell?

Likewise the authors model the Oag chain (O16) such that it is in some way captured by the opening at the top of the bell. Would it not be simpler if the Oag chain grows inside the bell and interacts with residues at the top of the bell when the bell is full?

One can speculate that this would then trigger a conformational changes in Wzz/Wzy that releases the polymer. This would be consistent with the most recent model and data from the the Lam group.

Overall this study greatly advances the field of bacterial polysaccharide biosynthesis research but needs to better address the published experimental data.

Other comments.

While the manuscript is well written in general, the Discussion is poorly structured. The first paragraph is highly irrelevant to the results of study, and while interesting needs to be introduced further down. The discussion is overly speculative and I have addressed key points above. While the authors discuss and critique Wzz/Wzy models, they do not actually present a model of their own, even though they have the best structure.

The scaffolding concept is similar to that suggested by Ref. 15.

I feel that the discussion is not accessible to anyone that even has a passing interest in the field. I feel the authors are overreaching and struggle with reconciling the vast amount of data on Wzz and related proteins. I suggest that they need to simplify the text to highlight and focus on their very important observations based on their wonderful structure.

p. 11. l. 14-15? Wzy proteins are highly hydrophobic and have few polar residues on the cytoplasmic side of the membrane. How can interaction with this Wzz region occur?

p. 13. l 5. WzzB is not needed for the stability of WzyB. The data cited (Nath et al) is based on mutant WzyB proteins, not the WT protein.

p. 13. l. 20-20. See above. wzz mutants do not have short Oag chains but unregulated ones.

p. 14. l. 16. The flower petal and pistil arrangement is the same as suggested in Ref. 15 (Fig. 3).

Minor.

p. l. l. 14. Looks more like a "box jellyfish" to me, not a squid!

p. 2. l. 10. "Gram negative"

p. 6. l. 11-12. This is incorrect. Reference 14 is the structure of mutant WzzB A107P from *S. flexneri*.

Reviewer #3 (Remarks to the Author):

This manuscript by Wiseman et al. reports a cryo-EM structure of the co-polymerase component (WzzB) of the Wzy-dependent polysaccharide biosynthesis pathway at an overall resolution of 3.2 Å. The structural study is performed on the full-length WzzB from *E. coli* and notably permits a high-resolution visualization of the transmembrane domain of Wzz family of proteins for the first time.

I have a few specific comments regarding the cryo-EM structure determination and the related analysis presented in the manuscript for consideration by authors:

- The authors did not observe intact WzzB particles for population 2 by cryo-EM analysis although this fraction corresponds to the main elution peak from the size-exclusion chromatography. I suggest that the authors carry out additional characterization, perhaps negative stain analysis, to better understand the behavior of the population 2.
- Related to the point above, what is the effect of protein concentration on the higher-order oligomerization (i.e. monomer/dimer species of the octameric assembly) of the WzzB.
- The 2D classes shown in the supplementary figure 2b seems to indicate a relatively faded density that likely corresponds to a second octamer. Is it possible that many of the particles that went into the high-resolution structure of the octameric complex correspond to the dimeric species on the micrograph? If so, authors can use an extensive classification scheme with a larger number of 3D classes to get a better separation of the particles.
- The proposed use of the Wzz periplasmic domain as a cryo-EM imaging scaffold for small membrane proteins is highly speculative in the absence of any experimental support. Furthermore, it is required that the small membrane protein anchored to the Wzz periplasmic domain is displayed as a rigid body to take advantage of the 8-fold symmetry. No data or convincing discussion is provided on how this might be achieved.

REVIEWER COMMENTS

We thank the reviewers for their comments and suggestions. Below we detail how we used the insightful comments to guide additional experiments and analysis to improve this manuscript.

Reviewer #1 (Remarks to the Author):

Several high resolution structures of the periplasmic segments of the bacterial polysaccharide co-polymerases, components of the O-antigen biosynthesis apparatus that are crucial in determining the length of the synthesized polysaccharide, have been determined over the last dozen years. However, the previous attempts by several labs to determine the structure of the full-length co-polymerase oligomer resulted in only low resolution structures, in which the transmembrane segments were not well resolved. Wieseman and colleagues succeeded in this endeavour and present in this manuscript the near-atomic resolution structure of the full-length E. coli polysaccharide co-polymerase determined by cryo-EM. Visualization of the transmembrane domains identified in addition to the two predicted TMs also a short N-terminal helix $\alpha 0$ aligned along the membrane surface that was not anticipated from the sequence. Together, this structure adds substantially to our understanding of co-polymerases and provides better starting point for hypothesis of how this structure controls the polymer length during biosynthesis.

Thanks, we agree!

While the structure shows for the first time the structural details of the transmembrane domains, the subsequent analysis of the structure could/should be much more thorough. The likely reasons why previous crystallographic and cryo-EM studies did not resolve the TMs, while the periplasmic domains were relatively well defined, is the flexibility of the linker regions between the TMs and the periplasmic segments and/or the effect of the detergent/solvent destabilizing the linkers. The helix bundles from different protomers are widely apart yet the authors do not speculate on what keeps them in the same relative orientation in every protomer. Are there small minicores within the linkers that provide some rigidity to the linkers? In what way Lys31 and Lys294 stabilize the two-helical bundles? What is the role of the N-terminal ~ 15 residues? Are they essential for the structural integrity of the oligomer? From the figures, it appears that they all come together in the center but are disordered. Would the construct with deletion of 15 N-terminal residues give similar cryo-EM particles? This would be a relatively easy experiment to perform.

We have created a construct with the N-terminal residues deleted and solved its structure; while producing a somewhat more compact dimer-of-octamers structure, the octameric structure is remarkably similar to the full-length structure including its TM domain. We have

also discussed in the text what we believe is stabilizing the TM domains. Please see the revised manuscript for details on these points.

There is a chapter on the cytoplasmic domain but since it was not modeled in the structure, there is really nothing to say about it in the result section. By the way, which residues were included in the model? The N-terminal residues shown in Figure 4 do not fit well the presented density.

Residues D19-K326 are included in the model. Similar to the L4 loop discussed below, the density around the N-terminal residues is at a lower resolution compared to the density of the TM helices. We chose a high threshold to show a good map-to-model fit of the transmembrane helices that creates the illusion of the first few residues being out of the density when at this high threshold level, at lower contouring levels the modeled N-terminal residues are clearly visible in the density.

Figure 1 leaves me with a question about modeling loop L4. In the C1 model there is a hole in the center of the oligomer indicating low density in this region. Where then the density in this region arrives from in the C8 model? Averaging weak density should still result in weak density.

We chose a high threshold that would clearly show that an octameric arrangement was obtained without any bias or the application of any symmetry in the C1 refinement. The figure was not intended to highlight the features of the L4 loops. Although slightly lower resolution than the bulk of the periplasmic bell, the L4 loops could still be built without difficulty and are visible at lower contouring levels of the map. The slightly lower resolution compared to the periplasmic bell would account for the appearance of a 'hole' when at very high threshold levels. We have updated the figure with a somewhat lower map contouring level to avoid this misunderstanding while still clearly showing an octameric arrangement in the C1 structure.

In Figures 2d and S2a I can see many double octamers, by my eye there more than 25% of them. Is there anything about the TM region from this reconstruction that show a similar arrangement of the TM helices? Does the thinner connection between the two octamers correspond to the cytosolic region or is it still part of the TM region? Superposition of the high-resolution octamer model on the low-resolution double octamer would be very useful and, hopefully, informative.

We have now performed this analysis. It is likely that the C-terminal chains from the 8xHis tag used for purification are interacting in the double octamers. Notably, we have been able to refine an octamer within the dimer to 3.7 Å and we do not see any difference between the two (see manuscript for details).

Discussion section should be reconsidered. It presents several hypotheses proposed from the structure of the model but there is no evidence at all presented in their support. The first chapter on a potential use of the periplasmic oligomer as a scaffold for creating larger assemblies of smaller transmembrane proteins for application in cryo-EM is interesting but has nothing to do with the result section and certainly should not be mentioned at the beginning of discussion, if at all. If there is data showing that this is indeed useful then it would be great.

Yes, we agree it is off topic and this section has been removed.

I do not see any new evidence from the structure reflecting on the mechanism of polysaccharide oligomers assembly and length control. The authors list models proposed in the literature but provide no data in support of any particular model. The discussion of a potential interaction between transmembrane segments of WzzB and Wzy polymerase is a bit of a hand waving with no data to support this. Many labs attempted to show physical interactions between WzzB and Wzy but although this is most likely, there is no direct evidence as yet of this association. It is possible that there might be 8 copies of Wzy around WzzB but the hypothesis of simultaneous synthesis of 8 polysaccharide chains around WzzB needs some evidence. WzzB presence leads to synthesis of polysaccharide chains that contain well over 20 repeating units of 4 sugars and the length of this polymer would exceed significantly the 100 Å distance from the bottom to the top of the periplasmic domain. So even if the tip of the polymer is locked by the L4 loop, what is the signal to stop adding new repeating units? In summary, I am really excited about the new structural details of the transmembrane segment of the co-polymerase WzzB but at the same time I would like to see a more in-depth analysis of the structure, in particular what keeps the TM aligned the same way in all protomers. Also, at least some data/evidence in support of the various hypotheses presented in the discussion.

We agree with the reviewer that these are still important and outstanding questions. Our intention with the discussion was to bring these up and formulate ideas for testable models for future experiments. We have revised this section, more clearly stating that these are speculations to be verified or rejected in future experiments.

Additionally Islam and Lam (Can. J. Microbiol. 697-716, 2014) do a wonderful job critiquing many of the various models in the literature. This reference is now included in the discussion.

Reviewer #2 (Remarks to the Author):

The authors describe the first full length structure of the O antigen (Oag) polysaccharide co-polymerase, WzzB of E. coli K12. Of particular significance is that they were able to determine at high resolution the structures of the TMs regions (TM1 and TM2), the tip region at top of the

periplasmic region/bell, and identify a short alpha helix ($\alpha 0$) at the amino terminal end of the protein. These were not visible in other structural studies. This new insight greatly helps to explain the impact of many mutagenesis studies targeting conserved residues that were previously published. The authors do not provide any new mutagenesis/LPS phenotype data based on their structure but interpret existing mutagenesis and cryoEM data in the context of their structure. While much of this is reasonable, aspects related to interaction of WzzB and with the Oag polymerase WzyB and Oag itself are highly speculative, and somewhat inconsistent with published data, and even their own WzzB structure. In several places in the Results and Discussion the experimental facts are misinterpreted and oversimplified.

The authors need to better define their terms such as less expression.

In the absence of WzzB, LPS with Oag is still produced but it lacks modal length control and hence has an unregulated chain length. WzyB is working quite well without WzzB.

Our intention was to state that, to our knowledge, no groups have been successful in attempts to purify Wzy, suggesting that it might require Wzz for stabilization during purification. We have corrected this in the text.

The authors speculate and model the location of the growing Oag polymer, and wrap it around the outside of the WzzB octamer. Is this energetically even feasible?

WzzB protein act on a diverse wide range of Oags (and a wide range of WzyB proteins). While they authors have used the O16 (as their WzzB comes from K12), this modelling needs to be done with different Oags to take into account this low specificity.

The O-antigen:WzzB interaction proposed is a model, to serve as a starting point to be confirmed or rejected in future studies, but at this stage additional modeling using other O-antigen structures is judged to be outside the scope of this study.

The authors mention the A107P mutant completely out of context when discussing TM interactions (p. 8. l. 8-11). What they fail to consider is the Chang et al determined that the A107P mutant has weaker binding to Oag, and that the A107 is located on the inner side of the WzzB dome. Furthermore, most mutations (in WzzB, FepE) affecting Oag chain length also map to the inner side of the dome. Their discussion and model ignore this data.

Thanks for the comment. We have removed the reference to the A107P on p8, and discussed it in a more fitting context in the discussion section along with the mutations affecting Oag length to the inside of the bell. Indeed, since there appears to be compelling evidence to support the Oag on both the inside and outside, we have also included a model with the Oag on the inside of the WzzB bell for discussion and comparison.

The authors have used a WzzB protein with a His₈ at the amino terminal end. Do they know if the protein is functional and confers a WT LPS phenotype?

We have not performed activity tests on our construct, however since our structure is consistent with the majority of the published biochemical data we believe our protein is functional. Also, the majority of the published biochemical data is with similar C or N terminal non-cleaved His-tagged protein that has been shown to be active; therefore, we have every reason to believe that our protein is also functional.

How does this poly His extension impact their interpretation of the base region / (α_0) in the cytoplasm?

We have created an N-terminally deleted WzzB and discussed this point in the text. However, due to the low resolution in this area of the protein, it is difficult to determine with confidence what its impact is.

The authors are unable to explain why the vast majority of the extracted WzzB is in a form that lacks a coherent structure. The authors study while consistent with other studies (i.e. octameric) fails to acknowledge that in all cases the most stable form of the protein is that which is used for structural studies. In vivo cross linking detects multiple forms of WzzB, so the evidence for the stable WzzB scaffold structure that authors model is weak. Indeed, the Wzz:WzyB interaction is likely to be weak based on published attempts to detect it. It is more likely that the WzzB self-interaction is dynamic in nature and hence structure.

Thank you for the comment! Yes, as structural biologists we are very much aware that the most stable form of the protein is the most often used for structural studies. To address this question, we have performed additional negative stain experiments on both populations and have discussed this point in the manuscript. Please see an updated Supplementary Figure 1, and the first paragraph of the results section.

The authors' excellent description of the orientation and function of conserved Gly residues in TM2 is inconsistent with their overall model for interactions between Wzz and Wzy. G312 is located inside the bell and is proposed to interact with Wzy. While taking this into account, their model places Wzy largely outside the bell. Would it not be simpler if Wzy is mostly inside the bell and polymerises an Oag that grows inside the bell?

This alternative model will be considered in future studies, but at the present time the precise location of the growing O-antigen cannot be given.

Likewise the authors model the Oag chain (O16) such that it is in some way captured by the opening at the top of the bell. Would it not be simpler if the Oag chain grows inside the bell and interacts with residues at the top of the bell when the bell is full?

To distinguish whether the non-reducing end of the O-antigen is 'captured' from within or from the outside cannot be determined just from the Wzz structure on its own, but needs additional information, to be obtained in futures studies of bacterial polysaccharide co-polymerase systems.

One can speculate that this would then trigger a conformational changes in Wzz/Wzy that releases the polymer. This would be consistent with the most recent model and data from the the Lam group.

Overall this study greatly advances the field of bacterial polysaccharide biosynthesis research but needs to better address the published experimental data.

Other comments.

While the manuscript is well written in general, the Discussion is poorly structured. The first paragraph is highly irrelevant to the results of study, and while interesting needs to be introduced further down.

Yes, we agree it is a bit off topic. We have removed this paragraph.

The discussion is overly speculative and I have addressed key points above. While the authors discuss and critique Wzz/Wzy models, they do not actually present a model of their own, even though they have the best structure.

The scaffolding concept is similar to that suggested by Ref. 15.

I feel that the discussion is not accessible to anyone that even has a passing interest in the field. I feel the authors are overreaching and struggle with reconciling the vast amount of data on Wzz and related proteins. I suggest that they need to simplify the text to highlight and focus on their very important observations based on their wonderful structure.

We have largely rewritten the discussion and tried to simplify it to focus the transmembrane domain.

p. 11. l. 14-15? Wzy proteins are highly hydrophobic and have few polar residues on the cytoplasmic side of the membrane. How can interaction with this Wzz region occur?

We have removed this comment.

p. 13. l 5. WzzB is not needed for the stability of WzyB. The data cited (Nath et al) is based on

mutant WzyB proteins, not the WT protein. Thanks! We have removed this reference from here.

p. 13. l. 20-20. See above. wzz mutants do not have short Oag chains but unregulated ones. We have adjusted the wording here and throughout the manuscript on this point.

p. 14. l. 16. The flower petal and pistil arrangement is the same as suggested in Ref. 15 (Fig. 3). We have removed this.

Minor.

p. 1. l. 14. Looks more like a “box jellyfish” to me, not a squid! We agree and have changed it accordingly.

p. 2. l. 10. “Gram negative” fixed!

p. 6. l. 11-12. This is incorrect. Reference 14 is the structure of mutant WzzB A107P from *S. flexneri*. Thanks! Removed ref 14 from here.

Reviewer #3 (Remarks to the Author):

This manuscript by Wiseman et al. reports a cryo-EM structure of the co-polymerase component (WzzB) of the Wzy-dependent polysaccharide biosynthesis pathway at an overall resolution of 3.2 Å. The structural study is performed on the full-length WzzB from *E. coli* and notably permits a high-resolution visualization of the transmembrane domain of Wzz family of proteins for the first time.

I have a few specific comments regarding the cryo-EM structure determination and the related analysis presented in the manuscript for consideration by authors:

- The authors did not observe intact WzzB particles for population 2 by cryo-EM analysis although this fraction corresponds to the main elution peak from the size-exclusion chromatography. I suggest that the authors carry out additional characterization, perhaps negative stain analysis, to better understand the behavior of the population 2.

Thanks for the suggestion, we have indeed performed a negative stain analysis on both populations. Please see an updated Supplementary Figure 1, and the first paragraph of the results section.

- Related to the point above, what is the effect of protein concentration on the higher-order oligomerization (i.e. monomer/dimer species of the octameric assembly) of the WzzB.

During cryo-EM grid screening and optimization various protein concentrations were tested. During the screening we did not notice a difference in the oligomerization between the various

protein concentrations in the cryo-EM test images. However, we did not explore this in detail as this is a non-physiological oligomerization and was not the aim of this study.

- The 2D classes shown in the supplementary figure 2b seems to indicate a relatively faded density that likely corresponds to a second octamer. Is it possible that many of the particles that went into the high-resolution structure of the octameric complex correspond to the dimeric species on the micrograph? If so, authors can use an extensive classification scheme with a larger number of 3D classes to get a better separation of the particles.

Yes, we have performed additional 2D and 3D classifications on the final set of particles and have indeed found additional dimers mixed in. Removal of these particles and re-refinement did not change the overall structure. Even being very strict with particle selection (ie removal of over 50 % of the particles) did not change the overall structure. Additionally, we created a mask around one of the octamers in the dimer and refined it to high resolution (see supplementary figure 3, right side of the processing tree). The resulting structure was the same as an individual octamer suggesting that both structures are equivalent. Thus, with this additional info that both states are equivalent, we feel that having a mixture of octamers and dimers-of-octamers in the final particle set doesn't matter for the final structure.

- The proposed use of the Wzz periplasmic domain as a cryo-EM imaging scaffold for small membrane proteins is highly speculative in the absence of any experimental support. Furthermore, it is required that the small membrane protein anchored to the Wzz periplasmic domain is displayed as a rigid body to take advantage of the 8-fold symmetry. No data or convincing discussion is provided on how this might be achieved.

As mentioned above, yes, it is a bit off topic. We thought it was interesting idea. We have removed this paragraph. Perhaps we can explore this idea in a separate future study.

REVIEWERS' COMMENTS

Reviewer #1 (Remarks to the Author):

The revised manuscript has been substantially improved and concentrates on the new findings of the membrane bound TMs. My comments were adequately addressed. The discussion is now to the point and provides new ideas about the role of the L4 loop and the interpretation of the mutational data on the outside and inside of the octamer.

I noticed two minor errors:

p. 6, l. 5 from the bottom - I believe that the authors refer to ref 13, not 14.

p. 14, l. 10 from the bottom - Figure 6, not 5.

Reviewer #2 (Remarks to the Author):

The authors have addressed my concerns and have made major improvements to the manuscript, and increased its accessibility to a broader audience.

Minor correction.

p. 14 l. 19. Fig. 5 should be Fig. 6.

Reviewer #3 (Remarks to the Author):

I am satisfied with the revised manuscript and the responses from authors.

NCOMMS-20-23700A

REVIEWER COMMENTS

Reviewer #1 (Remarks to the Author):

The revised manuscript has been substantially improved and concentrates on the new findings of the membrane bound TMs. My comments were adequately addressed. The discussion is now to the point and provides new ideas about the role of the L4 loop and the interpretation of the mutational data on the outside and inside of the octamer.

I noticed two minor errors:

p. 6, l. 5 from the bottom - I believe that the authors refer to ref 13, not 14. **-Thanks! Fixed.**

p. 14, l. 10 from the bottom - Figure 6, not 5. **-Thanks! Fixed.**

Reviewer #2 (Remarks to the Author):

The authors have addressed my concerns and have made major improvements to the manuscript, and increased its accessibility to a broader audience.

Minor correction.

p. 14 l. 19. Fig. 5 should be Fig. 6. **Thanks! Fixed.**

Reviewer #3 (Remarks to the Author):

I am satisfied with the revised manuscript and the responses from authors. **Great! Thanks!**